# The Role of Chemotherapy in Patients with Synchronous Colorectal Liver Metastases: A Nationwide Study

**DOI:** 10.3390/cancers17060970

**Published:** 2025-03-13

**Authors:** Hanna Sternby, Farima Brandt, Srinivas Sanjeevi, Jon Unosson, Souheil Reda, Carolina Muszynska, Jozef Urdzik, Petter Frühling

**Affiliations:** 1Department of Surgery, Institution of Clinical Sciences, Lund University, 221 84 Lund, Sweden; hanna.sternby@med.lu.se; 2Department of Surgical Sciences, Uppsala University, 751 85 Uppsala, Sweden; farima.brandt@uu.se (F.B.); srinivas.sanjeevi@uu.se (S.S.); jon.unosson@uu.se (J.U.); souheil.reda@uu.se (S.R.); carolina.muszynska@uu.se (C.M.); jozef.urdzik@uu.se (J.U.)

**Keywords:** colorectal cancer, liver metastases, metastasectomy, liver surgery, peri-operative chemotherapy, neoadjuvant chemotherapy, adjuvant chemotherapy

## Abstract

The role of peri-operative chemotherapy in patients with colorectal cancer and synchronous (upfront) resectable liver metastases is still under debate. One advantage of giving neoadjuvant chemotherapy is the idea of ‘test of tumor biology’. That is, to allow a period of time after chemotherapy to assess whether the metastases progress despite treatment. Arguably, this may preclude futile surgery. Neoadjuvant treatment may also downsize tumors, which can lead to a smaller liver resection (parenchymal-sparing surgery) and hopefully improve survival. On the other hand, a substantial number of patients experience adverse events during the course of treatment, and chemotherapy may lead to an increase in post-operative complications. The aim of this nationwide study was to assess long-term oncological outcomes of patients receiving neoadjuvant and adjuvant chemotherapy compared to those who underwent upfront hepatectomy.

## 1. Introduction

The role of peri-operative chemotherapy in patients with upfront resectable synchronous colorectal cancer with liver metastases (sCRLM) is not clear. The most common location for metastasis is the liver [1]. Chemotherapy, in combination with surgery may both reduce the risk of relapse and improve overall survival [2,3,4]. The treatment of synchronous CRLM is complex, and the criteria of R0-resectability of CRLM depend on technical and prognostic criteria together with the experience of the multidisciplinary team conference [5]. Technically, resectability is not dependent on tumor size or the number of bilobar involvements, as long as the hepatectomy can be performed leaving sufficient future liver remnant (≥30% future liver remnant). Prognostic factors include factors that influence disease-free survival or the likelihood of prolonged overall survival; these include metachronous/synchronous disease, the extent of liver metastases, and concomitant extrahepatic spread. In addition, mutational status, such as RAS, BRAF, or dMMR/MSI may influence overall survival and the treatment strategy [5,6]. Synchronous CRLM has less favorable cancer biology, and some recommend neoadjuvant chemotherapy unless resection of the primary cancer and liver metastases are considered easy [7].

Today, three randomized controlled trials exist [8,9,10,11] that have examined the role of chemotherapy in patients with resectable CRLM. In the EORTC Intergroup Trial 40983 [8] an improved progression-free survival was seen in the peri-operative chemotherapy group with 8.1% at 3 years compared to surgery alone. However, at a median follow-up of 8.5 years no difference could be discerned between the peri-operative chemotherapy group versus the surgery alone group in terms of overall survival [9].

As a rational extension of the EORTC trial—also known as the EPOC trial—another trial, the so-called, new EPOC trial, randomized patients with *KRAS* exon 2 wild-type resectable CRLM in a 1:1 ratio to receive chemotherapy with or without cetuximab before and after liver resection [12]. This trial was closed at the interim analysis since it met the protocol-defined futility criteria since the addition of cetuximab shortened the progression-free survival [11]. In a more recent randomized trial (JCOGO603), a modified infusional fluorouracil, leucovorin, and oxaliplatin (FOLFOX6) was added to liver surgery for liver-only metastatic colorectal cancer. This study demonstrated an improved disease-free survival at 5 years (50% versus 39%, *p* = 0.006) in the chemotherapy group, but there was no difference in overall survival [10]. Notably, in the EPOC trial 63% (n = 236) of the patients had metachronous liver metastases, whereas in the JCOGO603 trial 44% (n = 133) had metachronous disease. In Sweden, the standard of care has been the treatment plan described in the EORTC Intergroup Trial 40983 (EPOC trial), which includes six cycles of fluorouracil, leucovorin, and oxaliplatin before and after surgery.

In our previous study, neoadjuvant chemotherapy did not confer an improved overall survival in patients with a solitary colorectal liver metastasis [13]. The aim of the present study is to examine the role of neoadjuvant and adjuvant chemotherapy in patients with synchronous colorectal liver metastases (sCRLM) on overall survival.

## 2. Materials and Methods

### 2.1. Patients

This nationwide population-based cohort study comprised consecutive patients who underwent a resection for sCRLM and were registered in the National Quality of Registry for Liver, Bile Duct, and Gallbladder cancer (SweLiv) between 1 January 2009 and 31st of December 2017. To obtain information about the primary cancer, data were also retrieved and cross-linked with the Swedish Colorectal Cancer Registry (SCRCR).

### 2.2. National Quality Registries

Both SweLiv and SCRCR are prospectively maintained quality registries that contain data about the primary colorectal cancer, surgical, oncological, and interventional results. SweLiv was established in 2008 and is continuously updated against the Total Population Register at Statistics Sweden, which provides information regarding whether a patient is dead or alive. Both SweLiv and SCRCR have excellent coverage rates when compared with the Swedish Cancer Registry [14,15]. Data that were extracted included demographics, and inter alia information about the primary cancer diagnosis (primary cancer T stage, lymphatic spread of primary cancer), type of liver resection (major or minor), complications according to Clavien–Dindo classification, and information about neoadjuvant and adjuvant treatment.

### 2.3. Work-Up of Patients

All patients were discussed in colorectal- and liver-specific inter-disciplinary team conferences prior to treatment. Synchronous liver metastases were defined as the presence of metastases at the time of diagnosis or detected during staging of the primary cancer diagnosis. Patients were divided into two groups: those who had upfront surgery without neoadjuvant treatment and those who had neoadjuvant treatment prior to liver surgery.

### 2.4. Permissions

The study was approved by the regional ethical board (DnR 2019-00016).

### 2.5. Statistics

Continuous data are presented with median and interquartile ranges (IQRs). Categorical data are presented with proportions and percentages. Categorical variables were compared using a chi square test or Fisher’s exact test and continuous variables with the Mann–Whitney-U test. Overall survival was calculated from the date of surgery until death from any cause or lost to follow-up. Overall survival between the groups was visually compared with Kaplan–Meier curve, and any difference assessed with the log-rank test. In the multivariable Cox regression analyses, backward elimination was used. In the univariable and multivariable analyses, hazard ratios (HRs) are presented with 95% confidence intervals (CIs). Statistical significance was set as a two-sided *p* < 0.05. Data analyses were performed using SPSS IBM statistics Version 28, 2021.

## 3. Results

A total of 4943 patients were assessed for eligibility. After excluding metachronous CRLM and patients who had been treated with ablation, 2072 patients with sCRLM were included in the analysis (Figure 1). In total, 1238 patients (60%) were first treated with neoadjuvant chemotherapy before liver surgery. Forty percent (n = 834) had upfront surgery. In the neoadjuvant group, 755 patients (61%) had adjuvant treatment compared to 245 patients (29%) in the upfront surgery group.

Median age at surgery was 65 years (IQR 57–70 years) in the neoadjuvant group compared to 70 years (IQR 60–71 years) in the upfront surgery group (*p* ≤ 0.001) (Table 1). Nearly fifty percent (n = 404, 48%) were older than 70 years in the upfront surgery group compared to 28% (n = 343) in the neoadjuvant group (*p* < 0.001). In both groups the majority of patients were men. Most patients in both groups belonged to ASA classification 2 or 3 (80% in the neoadjuvant group compared to 84% in upfront surgery); a higher proportion of patients in the upfront surgery group belonged to ASA classification 4 (*p* = 0.006). Differences were discerned in the two groups in terms of T category of primary cancer (*p* < 0.027), lymphatic spread of primary cancer (*p* ≤ 0.006), tumor diameter (*p* = 0.019), and number of liver metastases (*p* ≤ 0.001). In the neoadjuvant group, 26% (n = 284) compared to 32% (n = 192) in the upfront surgery group had a T4 primary cancer, and 72% (n = 805) compared to 67% (n = 394) had a N1–N2 lymphatic spread of primary cancer. In the neoadjuvant group, 610 patients (51%) had three or more liver metastases compared to 263 patients (36%) in the upfront surgery group (Table 1).

### 3.1. Surgical Results and Morbidity

In both groups 12% were treated with an emergency operation because of the primary cancer (Table 2). In the neoadjuvant group, 392 patients (32%) compared to 141 patients (17%) had a rectal cancer. In the upfront surgery group, 23% (n = 194) of the patients had a right colectomy compared to 19% (n = 232) in the neoadjuvant group (*p* ≤ 0.001). Forty-three percent (n = 527) in the neoadjuvant group had a major hepatectomy (three or more liver segments) compared to 6% (n = 50) in the upfront surgery group (*p* ≤ 0.001). In the neoadjuvant group, 39% (n = 481) had a complete or partial response on the neoadjuvant chemotherapy treatment prior to surgery. In the neoadjuvant group, a radical resection (R0) was obtained in 81% (n = 898) compared to 80% (n = 298) in the upfront surgery group (*p* = 0.390). Rectal resection and left colectomies were more common in the neoadjuvant group (Table 2). Radical resection was defined as a resection margin of at least 1 mm. More patients in the neoadjuvant group experienced a complication graded as Clavien–Dindo 3a or greater compared to the upfront surgery group (*p* < 0.001). Eighty-one patients (7%) in the neoadjuvant group were admitted within 30 days compared to 36 patients (4%) in the upfront surgery group (*p* ≤ 0.001).

### 3.2. Overall Survival and Cox Regression Analysis

In the univariable analysis, several factors negatively influenced overall survival, such as advanced age (above 70 years), ASA classification 3 or 4, T3–T4 category, lymphatic spread of primary cancer, number of liver metastases, and upfront surgery (Table 3a). In the Kaplan–Meier analysis, there was a clear difference in overall survival between the neoadjuvant group compared to the upfront surgery group (Figure 2). The median overall survival in the neoadjuvant group was 57 months (95% CI 53–61 months), compared to 26 months (95% CI 23–29 months) in the upfront surgery group (log rank *p* ≤ 0.001).

In the multivariable analyses, however, neoadjuvant treatment (HR 1.04, 95% CI 0.86–1.26) did not confer an overall survival advantage. Adjuvant treatment was protective and prolonged overall survival (HR 0.80, 95% CI 0.69–0.94). The number of liver metastases strongly influenced overall survival, with more than six liver metastases doubling the risk of dying compared to one metastasis (HR 2.05, 95% CI 1.38–3.01). Moreover, several factors negatively influenced overall survival, including lymphatic spread (HR 1.68, 95% CI 1.41–1.99) and T3–T4 category of the primary cancer (HR 1.41, 95% CI 1.09–1.84), and age above 70 years (HR 1.46, 95% CI 1.25–1.70) (Table 3a).

### 3.3. Sub-Group Analyses

Since the multivariable Cox regression analyses clearly indicated that adjuvant treatment, as opposed to neoadjuvant treatment, conferred improved overall survival, a Kaplan–Meier curve was performed comparing all patients who had received adjuvant treatment (Figure 3). In this graph, we can see that the median overall survival improved in both the upfront surgery group and the neoadjuvant group and that there still was a difference in survival (44 months compared to 65 months, log rank test *p* ≤ 0.002). Visually, a greater improvement was made in the upfront surgery group, with an improvement of 18 months compared to 8 months in the neoadjuvant group.

To test the hypothesis that neoadjuvant treatment is especially important for patients with extensive liver metastases, we examined its role in patients with six or more liver metastases. In the multivariable analysis of this sub-group of patients (Table 3b), neither neoadjuvant treatment (HR 1.53, 95% CI 0.67–3.53) nor adjuvant treatment (HR 1.06, 95% CI 0.72–1.54) improved overall survival. In this analysis, advanced age (HR 1.93, 95% CI 1.30–2.87), ASA classification 3 or 4 (HR 1.58, 95% CI 1.06–2.36), and T3–T4 category of primary cancer of T3–T4 (HR 3.30, 95% CI 1.53–7.11) played an important role in overall survival.

### 3.4. Supplemental Analysis

To further examine the role of neoadjuvant and adjuvant chemotherapy on overall survival in sCRLM. We performed a multivariable analysis with all patients who had received adjuvant treatment (Appendix A). In this subgroup of patients, advanced age (≥70 years), lymphatic spread of primary cancer (N1–N2), and number of liver metastases (two or more metastases) negatively influenced overall survival. Neoadjuvant treatment (HR 1.04, 95% CI 0.81–1.34) and T category of primary cancer influenced overall survival in the univariable analysis but not in multivariable analysis. We also examined the role of chemotherapy in all patients who had received neoadjuvant treatment (Appendix A). In this group, advanced age (≥70 years), lymphatic spread of primary cancer, T category of primary cancer, and more than two liver metastases conferred worse overall survival. Notably, adjuvant treatment also positively prolonged overall survival (HR 0.82, 95% CI 0.68–0.97).

## 4. Discussion

Treatment strategies for patients with synchronous CRLM include a staged approach (either liver-first or colorectal-first) or a simultaneous approach [16]. In the simultaneous approach, both the liver and the colorectal resections are performed at the same time. In a staged approach, either the primary cancer is resected first (colorectal-first) or the liver metastases, with a period of recovery between the operations. In the only randomized controlled study, which compared colorectal-first with a simultaneous operation, no differences were found in complication rates [17]. Arguments for the colorectal-first approach include resection of the source of the metastases (the colorectal cancer), alleviation of potential bleeding, and reduced risk of obstruction or perforation of the primary cancer. And since many of these patients often commence with a course of chemotherapy, this may identify those patients who progress despite systemic treatment and may not benefit from a liver resection. Arguments for the liver-first approach include resection of the liver metastases, which is the most important determinant for prognosis. Moreover, since many of these patients have extensive metastatic involvement where resectability is borderline and made possible thanks to neoadjuvant treatment, this approach is thought to make use of the window of opportunity for curability [7,18,19,20].

The role of neoadjuvant chemotherapy in patients with resectable sCRLM is not clear. The present nationwide population-based study between 2009 and 2017 aimed to assess the influence of neoadjuvant and adjuvant chemotherapy on overall survival in sCRLM. In the univariable analysis, there was a survival advantage in the neoadjuvant group compared to upfront surgery. In the multivariable analyses, however, no survival benefit was seen in the neoadjuvant group. By contrast, adjuvant treatment appears to confer a survival benefit in the multivariable analyses.

Patients in the neoadjuvant group were younger (median age 65 compared to 70 years) and healthier since fewer patients belonged to ASA classification 3 and 4. A greater proportion in the upfront surgery group (32%, n = 192) had a T4 primary cancer compared to 26% (n = 284) in the neoadjuvant group (*p* = 0.027). By contrast, 72% (n = 805) in the neoadjuvant group compared to 67% (n = 394) in the upfront surgery group belonged to N1 or N2 category of the primary cancer (*p* = 0.006). Unfortunately, we only have data from patients who underwent liver surgery. An intention-to-treat analysis is therefore not possible since we cannot report on the number of patients who were treated with neoadjuvant treatment with the intention of receiving liver surgery but due to unforeseen reasons, such as progression of the disease or adverse events of the chemotherapy treatment.

Despite a vast amount of discussion in the literature, there is no consensus on the administration of chemotherapy and its ideal timepoint in resectable CRLM [5,21,22]. To date, there are three randomized controlled trials [8,10,12] that assess the importance of chemotherapy in resectable CRLM. In the EORTC trial, 364 patient were randomized to either peri-operative chemotherapy (six cycles before and six cycles after surgery of FOLFOX4) [8]. In this study, 35% (n = 128) had synchronous CRLM, and they were allowed to have between one to four liver metastases. In fact, more than 50% percent had only one liver metastasis, and only 7% (n = 26) had four liver metastases. This is in comparison to the current study where 72% (n = 1483) of patients had three or more metastases. In the EORTC trial, an improved progression-free survival at 3 years of 8.1% was seen in the chemotherapy arm; however, no difference was discerned in overall survival after 8.5 years of follow-up [9]. In a more recent RCT, 300 patients were randomly assigned to either hepatectomy or a combination of hepatectomy and modified infusional fluouracil, leucovorin, and oxaliplatin (mFOLFOX6) [10]. Severe adverse events were reported in approximately half of the patients, and there was no difference in overall survival.

Mitry et al. performed a pooled analysis of two phase-three trials to assess the importance of chemotherapy after hepatectomy. A comparison was made between patients who received surgery alone (n = 140) and patients who were first treated with hepatectomy and thereafter received six months of adjuvant chemotherapy [23]. Adjuvant chemotherapy was associated with progression-free survival and overall survival in the multivariable analysis. Araujo et al. performed a systematic review and meta-analysis, which included three RCTs and two observational comparative studies [2]. This study performed forest plots both for recurrence-free survival and overall survival (surgery + chemotherapy versus surgery alone) and concluded that chemotherapy improved overall survival (HR 0.77, 95% CI 0.67–0.88) and recurrence-free survival (HR 0.71, 95% CI 0.61–0.83). In a recent multicenter retrospective observational study by Di Martino and colleagues, which included 252 patients between 2010 and 2015, it was concluded that peri-operative chemotherapy for resectable CRLM was beneficial compared to neoadjuvant in terms of overall survival and recurrence-free survival [24].

To assess the role of neoadjuvant therapy in patients with multiple liver metastases, we performed a sub-group analysis. In this analysis we examined the role of neoadjuvant treatment in all patients with six or more liver metastases. To our surprise, not even in this group did neoadjuvant treatment prolong overall survival in the multivariable analysis. The assessment of whether a patient would benefit from peri-operative chemotherapy is complex and requires risk stratification and individual assessment in a multidisciplinary team conference [25]. Patients with resectable CRLM with a low risk of recurrence are likely to benefit from upfront surgery, and the role of adjuvant treatment post-operatively is not always easy to determine. By contrast, patients with resectable disease with a high risk of recurrence may benefit from peri-operative chemotherapy since a course of neoadjuvant treatment may function as a *test of biology*, where patients who clearly progress during the course of treatment are likely not to benefit from surgery. Treatment decisions in metastatic colorectal cancer need to be influenced by a rigorous understanding of the unique tumor characteristics of the primary cancer and the metastasis, such as molecular profiling, presence or absence of microsatellite instability, extent of tumor involvement and previous treatment, as well as each patient’s underlying co-morbidity [26]. Patients with metastatic colorectal cancer are highly heterogeneous. With advancements in surgical technique, the definition of resectability is also becoming more elusive. Today, technically, resectability is not determined by number and extent of liver metastases or bilobar disease, as long as a sufficient future liver remnant is preserved (≥30% of remnant liver) [5]. We today perform more complex liver surgery on elderly patients and have acquired an improved understanding of the molecular biology of colorectal cancer [26,27]. We should therefore become better at stratifying patients into low–medium–high-risk groups and weigh this information against *oncological criteria*, concerning prognostic factors and the biology of the disease for each patient.

This study has several limitations. One is the fact that the registers do not allow us to differentiate between patients with primarily unresectable disease, who became resectable as a result of neoadjuvant treatment, from those who were resectable from the outstart. Although the definition of resectability is prima facie easy to define, what exactly constitutes resectable disease is highly influenced by local practices [28] and is constantly evolving. Also, information on an individual level regarding the selection between neoadjuvant treatment or upfront surgery is not available. Traditionally, neoadjuvant treatment has been the standard of care in the synchronous disease and upfront surgery mainly in the metachronous setting or if the patient was considered too frail for chemotherapy. Additionally, the registers do not allow us to provide detailed information on the type of chemotherapy or number of cycles [29,30]. On the other hand, the aim of the study was not to assess specific treatment responses or tolerability to oncological agents but to examine real-word data. Lastly, although the registers show high accuracy and validity, there was a degree of missing data for some variables. Another limitation of the present study is that we cannot determine from the register which patients had vanishing lesions and for what reasons patients would not have the whole treatment sequence (liver and colorectal resection). Given the retrospective design and observational nature of the study, there is a risk of unmeasured confounding variables. The aim of our study is to shed light on daily practice and to offer real-world data on the role of chemotherapy on overall survival. Since we do not have information regarding the choice of neoadjuvant/adjuvant treatment and number of cycles, we are unable to provide information about the tolerability and efficacy of specific chemotherapy agents.

One strength of the present study is its population-based design. Moreover, data obtained for the study come from national quality registers with high accuracy and validity. The long follow-up period entails that few patients have been censored, which adds stability to the overall survival analyses. Another advantage of the present study is that it focuses solely on synchronous CRLM, which makes the study population somewhat more homogenous. Although our study is a population-based study and there might be a risk of unmeasured confounding variables, we believe the results suggest a more cautionary approach in administrating peri-operative treatment to all patients with resectable sCRLM. This position is convincingly presented by Booth and Berry [31] when reflecting on the findings presented in the RCT by Kanemitsu et al. (JCOG0603) [10]. According to them, the JCOG0603 trial ‘signals’ the ‘end of an era’ in which peri-operative and postoperative chemotherapy can be viewed as the ‘default standard for patients with resectable CRLM’ [31]. Since chemotherapy may only delay recurrence, without any clear evidence that it prolongs survival, a six-month chemotherapy treatment period with frequent clinic visits and the risk of severe adverse events may be questioned.

## 5. Conclusions

To our knowledge, this is the largest population-based study to examine the role of chemotherapy on resectable synchronous CRLM and overall survival. In the present study, an association between adjuvant chemotherapy and overall survival was found; this was not the case for neoadjuvant treatment.

## Figures and Tables

**Figure 1 cancers-17-00970-f001:**
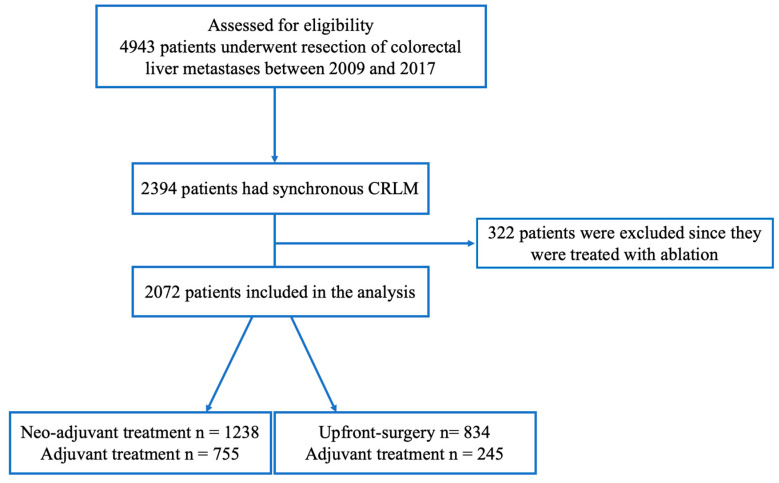
A flowchart of the study population.

**Figure 2 cancers-17-00970-f002:**
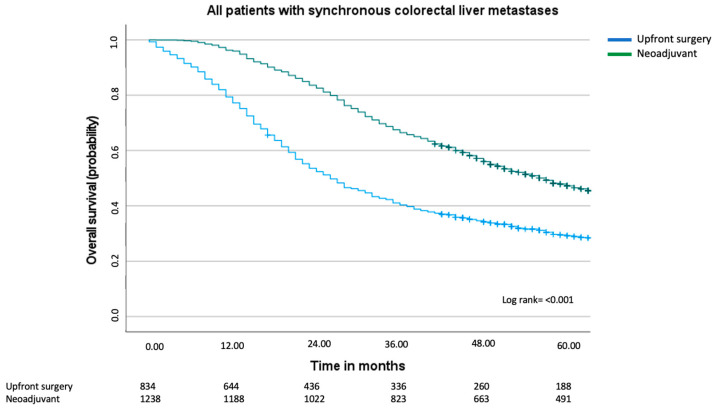
Overall survival including all patients with synchronous colorectal liver metastases. Median overall for the upfront surgery group was 26 months (95% CI 23–29 months) and for the neoadjuvant group, 57 months (95% CI 53–61 months), log rank = *p* < 0.001.

**Figure 3 cancers-17-00970-f003:**
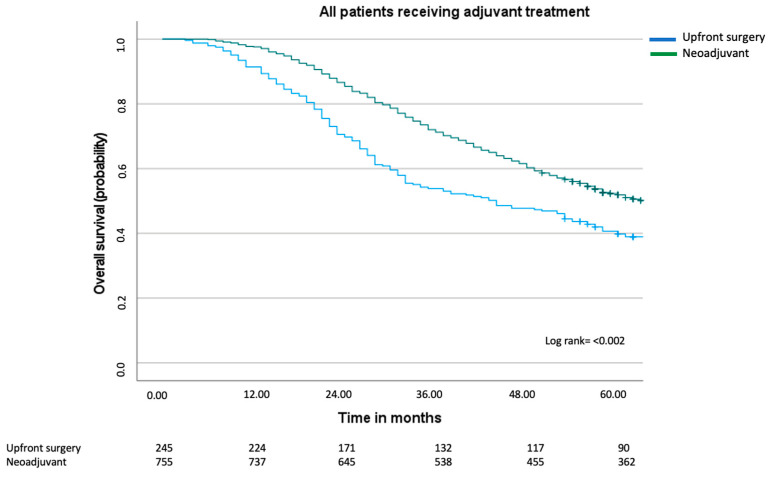
Overall survival including all patients with synchronous colorectal liver metastases who received adjuvant treatment. Median overall for the upfront surgery group was 44 months (95% CI 32–56 months) and for the neoadjuvant group 65 months (95% CI 58–72 months), log rank = *p* < 0.002.

**Table 1 cancers-17-00970-t001:** Baseline characteristics of study population.

	No. Patients	Neoadjuvant	Upfront Surgery	*p* Value
	n = 2072	n = 1238	n = 834	
**Age (years)**				
Median (IQR)	66 (58–72)	65 (57–70)	70 (60–71)	<0.001
Age > 70	747 (36)	343 (28)	404 (48)	<0.001
**Sex**				
Men	1269 (61)	765 (62)	504 (60)	0.282
Women	803 (39)	473 (38)	330 (40)	
**ASA**				
1	289 (17)	206 (19)	83 (14)	0.006
2	947 (57)	631 (57)	316 (55)	
3	419 (25)	249 (23)	170 (29)	
4	21 (1)	12 (1)	9 (2)	
Missing	396	140	256	
**T category of primary cancer**				
T1	21 (1)	13 (1)	8 (1)	0.027
T2	137 (8)	98 (9)	39 (7)	
T3	1045 (62)	692 (64)	353 (60)	
T4	476 (29)	284 (26)	192 (32)	
Tx/missing 592	393	151	242	
**Lymphatic spread of primary cancer**				
N0	503 (30)	308 (28)	195 (33)	0.006
N1	642 (38)	449 (40)	193 (33)	
N2	557 (32)	356 (32)	201 (34)	
Nx/missing	370	125	245	
**Tumor diameter (mm)**	20 (13–33)	20 (14–35)	20 (13–30)	0.019
Median (IQR)				
**Number of liver metastases**				
1	666 (35)	328 (28)	338 (47)	<0.001
2	372 (19)	249 (21)	123 (17)	
3–5	502 (26)	373 (31)	129 (18)	
6	253 (13)	181 (15)	72 (10)	
>6	118 (6)	56 (5)	62 (9)	
Missing	161	51	110	

**Table 2 cancers-17-00970-t002:** Details of primary cancer resection, hepatectomy, and surgical results.

	Neoadjuvant	Upfront Surgery	*p* Value
	n = 1238	n = 834	
**Emergency primary cancer operation**		99 (12)	0.059
	144 (12)		
**Type of colorectal resection**			
Rectal resection	392 (32)	141 (17)	<0.001
Left colectomy	423 (34)	195 (23)	
Right colectomy	232 (19)	194 (23)	
Colectomy, other	62 (5)	37 (4)	
Unspecified	117 (10)	225 (27)	
Laparotomy only	42 (1)	42 (5)	
**Major hepatectomy**			
3 or more liver segments	527 (43)	50 (6)	<0.001
**Intraoperative blood loss**			
**Median (IQR)**	600 (300–1100)	300 (125–475)	0.083
**Response on neoadjuvant treatment**			
Complete/partial	481 (39)		
Stable disease	98 (8)	N/A	
Progress	34 (3)		
Unclear	625 (50)		
**Liver resection**			
R0	898 (81)	298 (80)	0.39
R1	125 (11)	36 (10)	
Unclear	89 (8)	37 (10)	
Missing	126	463	
**Post-operative complications**			
Re-admission within 30 days	81 (7)	36 (4)	<0.001
Clavien Dindo 3a	87 (7)	25 (3)	<0.001
Clavien–Dindo 3b	41 (3)	16 (2)	
Clavien–Dindo 4a	15 (1)	5 (1)	
Clavien–Dindo 4b	4 (0)	2 (0)	
Clavien–Dindo 5	5 (0)	2 (0)	

**Table 3 cancers-17-00970-t003:** (**a**) Cox regression analyses and overall survival. (**b**) Cox regression model for overall survival for patients with six or more than six liver metastases.

(a)					
	Univariable	*p* Value		Multivariable	*p* Value
	HR, 95% CI			HR, 95% CI	
**Age (years)**					
<70	Reference			Reference	
≥70	1.38 (1.26–1.56)	<0.001		1.46 (1.25–1.70)	0.007
**Gender**					
Women	Reference			Reference	
Men	1.05 (0.90–1.11)	0.931		1.03 (0.89–1.19)	0.719
ASA					
1–2	Reference			Reference	
3–4	1.26 (1.09–1.46)	0.002		1.16 (0.98–1.38)	0.087
**T category of primary cancer**					
T1–T2	Reference			Reference	
T3–T4	1.91 (1.53–2.39)	<0.001		1.41 (1.09–1.84)	0.010
**Lymphatic spread of primary cancer**					
N0	Reference			Reference	
N1–N2	1.89 (1.64–2.16)	<0.001		1.68 (1.41–1.99)	<0.001
Number of liver metastases					
1	Reference				
2	1.33 (1.13–1.56)	<0.001		1.52 (1.26–1.84)	<0.001
3–5	1.40 (1.21–1.63)	<0.001		1.38 (1.14–1.66)	<0.001
6	1.80 (1.51–2.15)	<0.001		1.54 (1.22–1.96)	<0.001
>6	3.04 (2.44–3.79)	<0.001		2.05 (1.38–3.01)	<0.001
**Chemotherapy**					
Upfront surgery	Reference				
Neoadjuvant	0.56 (0.51–0.63)	<0.001		1.04 (0.86–1.26)	0.681
No adjuvant	Reference				
Adjuvant	0.62 (0.55–0.69)	<0.001		0.80 (0.69–0.94)	0.007
(**b**)					
	**Univariable**	***p* Value**	**Multivariable**		***p* Value**
	**HR, 95% CI**		**HR, 95% CI**		
**Age (years)**					
<70	Reference		Reference		
≥70	1.61 (1.25–2.07)	<0.001	1.93 (1.30–2.87)		0.002
**Gender**					
Women	Reference		Reference		
Men	0.99 (0.78–1.25)	0.928	1.21 (0.82–1.76)		0.335
ASA					
1–2	Reference		Reference		
3–4	1.33 (0.95–1.86)	0.094	1.58 (1.06–2.36)		0.026
**T category of primary cancer**					
T1–T2	Reference		Reference		
T3–T4	3.81 (1.87–7.73)	<0.001	3.30 (1.53–7.11)		0.002
**Lymphatic spread of primary cancer**					
N0	Reference		Reference		0.241
N1–N2	1.94 (1.33–2.83)	<0.001	1.34 (0.82–2.17)		
**Chemotherapy**					
Upfront surgery	Reference		Reference		
Neoadjuvant	0.301 (0.24–0.38)	<0.001	1.53 (0.67–3.53)		0.314
No adjuvant	Reference		Reference		
Adjuvant	0.70 (0.54–0.90)	0.005	1.06 (0.72–1.54)		0.778

## Data Availability

The data may be available from the corresponding author based on reasonable request.

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
