# Peer review of "The Role of Chemotherapy in Patients with Synchronous Colorectal Liver Metastases: A Nationwide Study"

_cancers, 2025, doi:10.3390/cancers17060970_

Round 1

Reviewer 1 Report

Comments and Suggestions for Authors

A more detailed presentation of the current guidelines regarding metastatic colon cancer would be useful, a fact that would give greater coherence to the article and emphasize the importance and complexity of the therapeutic decision presented in this study.

Congratulations for the work.

Reviewer 2 Report

Comments and Suggestions for Authors

Thank you for giving me the opportunity to review this population cohort study based on registry data. It is a real-life observational study which differs from studies from expert centres.

In this respect, I don't really understand how the patients were initially treated, hence this question about the rest of the study. If the metastases were synchronous liver metastases, when was the surgery on the primary colon performed? did the patients in the primary surgery arm have combined colorectal and liver surgery, and those in the neo-adjuvant arm chemotherapy and/or radiotherapy followed by combined surgery?

because in the case of resection of the primary colorectal site, how many patients have had adjuvant chemotherapy before resection of the liver metastases?

With regard to the comparability of the groups, perhaps a propensity score should have been used, given the number of patients, in order to increase the statistical power of the results. 

Reviewer 3 Report

Comments and Suggestions for Authors

 The present study investigates the oncological impact in terms of survival of chemotherapy in patients resected for colorectal cancer and synchronous liver metastases, particularly the role of neoadjuvant chemotherapy. At the same time, potential predictors for the long-term survival are explored. The study is a nationwide level one and includes many patients. The present study would interest the journal readers because the approached topic is fascinating to clinical practice. Thus,  nowadays, there are different approaches worldwide regarding the use of neoadjuvant chemotherapy for such patients, and the previously published data addressing the issue of neoadjuvant chemotherapy in colorectal cancer patients with synchronous liver metastases reached conflicting results regarding the potential benefit for long-term survival. At the same time, concerns were previously raised about the possible detrimental effects of neoadjuvant chemotherapy for such patients: increased rates of complications, particularly for liver resections and vanishing metastases. The present paper mainly highlights the increased median overall survival after neoadjuvant therapy compared to upfront surgery chemotherapy in patients resected for colorectal cancer and synchronous liver metastases, albeit the neoadjuvant chemotherapy was not identified as an independent predictor of overall survival. Thus, better survival rates in the neoadjuvant chemotherapy compared to the upfront surgery group of patients might be due to factors other than the neoadjuvant approach (see below). Adverse prognostic factors for overall survival were identified as being older age, T stage of the colorectal tumor, positive lymph nodes of the colorectal tumor, and number of liver metastases. Notably, the use of adjuvant therapy was identified as a favorable prognostic factor for overall survival. Thus, the present study may bring a few novelties to the field and clinical practice, albeit a few identified prognostic factors were previously described in other studies. While the manuscript is relatively well-designed and written, a few methodological flaws may alter the results and conclusions.

The present study's strength is the large number of analyzed patients in a nationwide populational-based study. At the same time, the weak points are the retrospective design and the relatively long period of inclusion, which may reflect different chemotherapy regimens/practices. Furthermore, the issues of futile surgery, vanishing metastases, liver and/ or colorectal surgery-related complications, and progression of disease under neoadjuvant chemotherapy that precludes further resection of the primary colorectal tumor and/ or liver metastases were not tackled. Nevertheless, the criteria for allocation to upfront/ neoadjuvant chemotherapy are not provided.

A few issues should be addressed before considering the acceptance:

Major issues:

The study shows that in the upfront surgery group, patients were significantly older and had lower adjuvant therapy rates compared with the neoadjuvant therapy group. Furthermore, significant differences between the groups were also observed for ASA scores, T and N stages of the colorectal tumor, tumor diameter, and number of liver metastases. At the same time, negative prognostic factors for overall survival were identified as being older age, T stage of the colorectal tumor, positive lymph nodes of the colorectal tumor, and number of liver metastases; a favorable prognostic factor was adjuvant chemotherapy.  Nevertheless, other factors, such as the location of the primary tumor (left vs. right colon, colon vs. rectum), mutation status, type of hepatectomy, complications rates,  time to adjuvant chemotherapy after resection, type of adjuvant chemotherapy or neoadjuvant therapy regimens/ completion of such therapies might influence the overall survival; a few of these potential factors were not even considered in the comparative analyses between the groups while a few other factors significantly differed between the analyzed groups of patients. Considering the differences between the groups for important predictors of survival, please consider using a propensity score analysis for the overall survival adjusted for significant predictors that might influence survival in the two groups. Thus, the present study's results and conclusions might change or be sustained by more substantial analyses.

Please consider addressing a few issues (related to the current cohort of patients), such as criteria for allocation to upfront/ neoadjuvant chemotherapy (if available), futile surgery, vanishing metastases, liver and/ or colorectal surgery-related complications, and progression of disease under neoadjuvant chemotherapy that precludes further resection of the primary colorectal tumor and/ or liver metastases were not tackled.

It appears adjuvant therapy was more frequently used in the neoadjuvant group (61%) compared to only 29% of patients in the upfront surgery. This aspect might significantly explain differences in overall survival in the upfont surgery vs neoadjuvant group of patients. Please explain the important differences between the two groups of patients when using adjuvant therapy.

Minor issues:

Please consider including a few limitations of the study, as mentioned above.

Considering the results of the present study, how do the authors consider using neoadjuvant chemotherapy in clinical decision-making for such patients? It is worth mentioning that in the present cohort, 60% of patients underwent neoadjuvant chemotherapy despite weak evidence-based support.

Please consider referencing the statement from lines 63-65.

In Table 2, please consider providing the p-value for the emergency surgery.

How do the authors explain non-significant differences in intraoperative blood loss between the groups despite differences, at least for major liver resections?

Put the reference citations in the format required by the journal.

Lines 320-329 should make references to the supplemental material. 

Round 2

Reviewer 2 Report

Comments and Suggestions for Authors

I have no comments

Reviewer 3 Report

Comments and Suggestions for Authors

Unfortunately, the authors did not address a few significant issues, such as potential factors that might influence the oncological outcomes and results of the present study. Thus, there are still gaps in the methodology. Using propensity score matching analyses is strongly recommended.